# Computational Design of an Electro-Membrane Microfluidic-Diode System

**DOI:** 10.3390/membranes13020243

**Published:** 2023-02-17

**Authors:** Mykola Bondarenko, Andriy Yaroshchuk

**Affiliations:** 1F.D. Ovcharenko Institute of Bio-Colloid Chemistry, National Academy of Sciences of Ukraine, Vernadskiy ave.42, 03142 Kyiv, Ukraine; 2ICREA, pg. L.Companys 23, 08010 Barcelona, Spain; 3Department of Chemical Engineering, Polytechnic University of Catalonia–Barcelona Tech, av. Diagonal 647, 08028 Barcelona, Spain

**Keywords:** ion-exchange layer, nanoporous layer, electroosmosis, micro-perforation, net volume flow, current-voltage characteristics

## Abstract

This study uses computational design to explore the performance of a novel electro-membrane microfluidic diode consisting of physically conjugated nanoporous and micro-perforated ion-exchange layers. Previously, such structures have been demonstrated to exhibit asymmetric electroosmosis, but the model was unrealistic in several important respects. This numerical study investigates two quantitative measures of performance (linear velocity of net flow and efficiency) as functions of such principal system parameters as perforation size and spacing, the thickness of the nanoporous layer and the zeta potential of the pore surface. All of these dependencies exhibit pronounced maxima, which is of interest for future practical applications. The calculated linear velocities of net flows are in the range of several tens of liters per square meter per hour at realistically applied voltages. The system performance somewhat declines when the perforation size is increased from 2 µm to 128 µm (with a parallel increase of the inter-perforation spacing) but remains quite decent even for the largest perforation size. Such perforations should be relatively easy to generate using inexpensive equipment.

## 1. Introduction

Conventional diodes are electronic elements having strongly asymmetric current-voltage characteristics that are used primarily for current rectification. Many recent studies have explored the so-called nanofluidic diodes [1,2,3,4,5,6,7,8,9,10,11,12,13,14,15,16,17]. Despite the “fluidic” in the name, most of them actually did not consider fluid flows as such and dealt only with Ion Concentration Polarization (ICP) and current rectification in fluid-filled nanochannels or nanopores. Only a few studies actually studied the associated volume transfer, for example, in nano-funnels [18], or track-etched membranes with conical nano-pores [19,20,21]. However, the volume flows often were tiny, and scenarios of parallelization were not clear. On the other hand, Alternating Current Electro-Osmotic (ACEO) pumps based on non-linear and non-stationary electrokinetics (traveling-wave ACEO using asymmetric electrode arrays [22,23,24,25,26,27,28]) typically featured very low internal hydraulic resistances, so they were not suitable for pumping against any noticeable hydraulic resistances.

To be asymmetric, electroosmosis (EO) must be non-linear. One of the possible mechanisms of EO non-linearity is related to a concurrent ICP. Ion-Exchange (IEX) membranes are well-known to have non-linear and asymmetric current–voltage characteristics occurring due to current-induced ICP. However, their electro-osmotic permeability is very low, so in 1D configurations, they do not generate appreciable volume flows. Nevertheless, in microanalysis, for example, in the so-called T-junction configurations (typically using cation-exchange Nafion materials), coupling between ICP and “orthogonal” EO flows gives rise to interesting and useful analyte pre-concentration phenomena [29,30,31,32,33,34]. These systems demonstrate the potential of “non-1D” configurations. Effectively, such a configuration was introduced and explored (using numerical simulations) in a previous study [35], which studied a new kind of electro-membrane material combining current-induced ICP with significant EO flows. This occurred as a result of a physical conjugation of the IEX and nanoporous layers, while micro-perforations in the IEX layers were enabled through volume flows. Such structures featured considerable asymmetries in the rates of volume transfer (at the same current magnitude) depending on the current direction, were easily upscalable with increasing membrane area and, thus, potentially suitable for an electrically driven “power fluidics” (by analogy with power electronics).

The previous study [35] considered the simplest model already featuring the behavior of interest. Thus, for instance, and for simplicity, the electrolyte concentration was set right at the external surface of the IEX layer, so a perfect stirring of the outside solution was assumed. This assumption is unrealistic especially for the reversed voltage when this external interface becomes depleted due to the current passage. In addition, the previous study set a constant electrostatic potential right at the external surface of the ion-exchange layer. This was performed to avoid explicitly considering the external solution (which would represent a considerable numerical complication), but this condition actually was not compatible with the condition of a given electrolyte concentration because the condition of the given potential would require putting a reversible electrode right at the external surface, while the condition of a given electrolyte concentration would demand perfect solution stirring right to the surface (and this would actually be impeded by the presence of any electrode). In addition to that, the previous study considered only opposite signs of fixed charges in the IEX and porous layers, only one perforation size and only one value of the zeta potential.

In this study, we will make the model essentially more realistic by relaxing several of those approximations. At the same time, due to this, the model parameters will become too numerous for a systematic parametric study to be feasible. Nonetheless, we will explore some correlations of potential interest for eventual practical applications. In particular, we will consider dependencies of system performance on the size of perforations and distance between them, on the thickness of the porous layer as well as the sign and magnitude of the zeta potential of the porous layer. All of these dependencies will reveal more or less pronounced maxima, which can be exploited in the future optimization of practical applications on a case-by-case basis. We will also see that considerable volume fluxes (in the range of several tens of liters per square meter per hour) can be expected at realistic parameter combinations.

In this study, we consider only stationary solutions, which implies that the electrical capacity of electrodes is sufficiently large to ensure practically constant voltage drops in the system over times that are much longer than the characteristic relaxation times of concentration changes. This assumption is probably not very realistic, so for future analyses of practical systems, non-stationary simulations will be needed.

## 2. Theory

We consider a porous layer coated on one side with an IEX “mask” having rather scarce circular perforations (see Figure 1). Strictly speaking, one should specify the pattern of the perforation array and solve 3D problems for volume and ion transfer. For infinite arrays, proceeding from symmetry considerations, one can consider a single perforation while accounting for the existence of the other ones using boundary conditions formulated at the external surface of a 3D domain enclosing every single perforation. Depending on the (regular) pattern, such domains can have triangular, square or hexagonal cross-sections. Their explicit modeling would require the solution of 3D problems. To reduce 3D problems to 2D, we will use a cylindrical cell model that approximates the polygonal domain surface with a cylinder. Several studies on microelectrodes [36,37,38] have demonstrated good accuracy of such an approach.

As shown in Figure 1, our model consists of four layers, namely, a porous layer, an IEX layer, an unstirred layer and a further solution layer (from top to bottom). In the IEX layer, there is a single circular perforation. Similar to the previous study, for simplicity, we will neglect a finite thickness of the IEX “mask” and consider it a geometrical boundary impermeable to volume flow and coions (see below).

For the description of liquid flow in the solution layers beneath the membrane, we use the Stokes and continuity equations
(1)ηΔv−∇p=0
(2)divv=0
where v is the vector fluid velocity, η is the dynamic viscosity, p is the hydrostatic pressure, Δ is the Laplace operator and ∇ is the nabla operator.

Pressure-driven flows through porous media are often described by the Darcy law, which postulates that the flow rate is proportional to the negative hydrostatic pressure gradient. However, the use of the Darcy equation gives rise to non-zero slip at non-porous surfaces and to non-physical singularities in the normal velocity at the perforation edge (see, for example, [35]). To avoid such singularities, we will use the so-called Brinkman equation [39], which is a kind of “superposition” of the Darcy and Stokes equations. Strictly speaking, this equation lacks a rigorous physical background. Nonetheless, it correctly described the limiting cases of Darcy and Stokes flows and is compatible with the no-slip condition at solid surfaces. The conventional Brinkman equation (for pressure-driven flows) reads as:(3)$$\frac{\eta }{\gamma }\Delta u-\nabla p-\frac{u}{\kappa }=0$$
where u is the vector cross-section-averaged fluid velocity, k is the porous-medium hydraulic permeability and γ is its porosity. As mentioned above, the Brinkman equation is a “superposition” of the Stokes and Darcy equations. These two equations use different kinds of velocity. While the Stokes equation operates with actual fluid velocity, the Darcy equation uses a cross-section-averaged fluid velocity. Dividing viscosity by porosity in the “Stokes component” of Equation (3) (the first term on the left-hand side) effectively gives rise to some reduction in velocity proportionally to the porosity and, thus, “recalibrates” the velocity to the cross-section-averaged one. As a result, both “components” of the Brinkman equation operate with the same kind of velocity.

As demonstrated in the previous study, in the so-called Smoluchowski approximation for the description of EO, the hydrostatic pressure gradient should be modified to include an electrical body force. As a result, the EO Brinkman equation will read as
(4)$$\frac{\eta }{\gamma }\Delta u-\nabla p+\frac{\beta }{k}\nabla \varphi -\frac{u}{\kappa }=0$$
where φ is the electrostatic potential, and the coefficient β is defined as
(5)$$\beta \equiv \frac{\varepsilon \varepsilon_{0}\zeta \gamma }{\eta }$$
where $\varepsilon \varepsilon_{0}$ is the liquid dielectric constant, ζ is zeta potential of the pore surface and γ is the active porosity (including pore tortuosity). Using Equation (5) implies that the pores are sufficiently large compared to the Debye screening length (see below for the applicability of this approximation).

Usually, this has to be complemented by the continuity equation.
(6)divu=0

In addition, we require equations for electrostatic potential and salt concentration. As such, we will use Ohm’s law and the convection–diffusion equation. They have the same form in both porous and unstirred layers (recall that we do not explicitly consider the “interior” of the IEX layer), the only difference being that electric conductivity and diffusion permeability are reduced in the porous layer due to the finite porosity. Thus, in the unstirred layer
(7)I=−αc∇φ
(8)$$J_{s}=-D\nabla c+cv$$
and in the porous layer
(9)$$I=-\alpha_{p}c\nabla \varphi$$
(10)$$J_{s}=-D_{p}\nabla c+cu$$
where I is the vector current density, $J_{s}$ is the vector salt flux, D is the salt diffusion coefficient and c is the salt concentration. For simplicity, in this study, we consider (1:1) electrolytes with cations and anions of the same mobility (such as in KCl). Therefore, the proportionality coefficient in Equation (7) (Ohm’s law) is given simply by
(11)$$\alpha \equiv \frac{2F^{2}D}{RT}$$

Finally, due to the finite porosity
(12)$$D_{p}\equiv D\gamma$$
(13)$$\alpha_{p}\equiv \frac{2F^{2}D_{p}}{RT}$$

Using the salt concentration in the porous layer, we can understand the concentration averaged over the pore space.

By using Ohm’s law, we can neglect the so-called streaming currents (convective transfer of electric charge due to the movement of charged pore liquid) as compared to conventional electromigration currents. This is legitimate for sufficiently large pores (compared to the screening length).

The equations to solve are those of charge and salt conservation, that is
(14)$$div\left(I\right)=0$$
(15)$$ddi\left(J_{s}\right)=0$$
which apply in each of the layers.

## 3. Boundary Conditions

### 3.1. External Boundary of the Porous Layer

Strictly speaking, there is also an unstirred layer. However, due to the relatively large pore size, this interface is not current polarized. Therefore, any unstirred layer can be effectively included in the porous layer. The boundary conditions here are zero hydrostatic pressure, zero electrostatic potential and the given salt concentration. Due to the problem of linearity in concentration, the concentration can be set at any level
(16)$$p\left(H\right)=0$$
(17)$$\varphi \left(H\right)=0$$
(18)$$c\left(H\right)=1$$

Since we are solving the Brinkman equation, we need an additional hydrodynamic boundary condition at this surface. As such, we used the following condition of flow one-dimensionality
(19)$$u_{\rho }\left(H\right)=0$$

### 3.2. External Surface of the Unstirred Layer

Here, we set the same value of the salt concentration and a non-zero applied voltage
(20)$$c\left(-L\right)=1$$
(21)$$\varphi \left(-L\right)=\varphi_{0}$$

Due to deviations from flow one-dimensionality close to the perforations, there are some hydrostatic pressure gradients in the solution beneath the membrane. They can be expected to fade away at some distance from it giving rise to constant hydrostatic pressure and a 1D liquid flow. In auxiliary simulations, we checked the dependence of our results on the location of the plane where hydrostatic pressure was set equal to zero, and we found that there was no dependence at $L_{s}≅400÷500\;\mu m$. Given that there is no applied pressure difference in our system (EO conditions),
(22)$$p\left(-L_{s}\right)=0$$
(23)$$v_{\rho }\left(-L_{s}\right)=0$$

The hydrodynamic equations are solved within the whole layer from $-L_{s}$ up to the membrane surface. However, the salt-transfer equations are solved only within the unstirred layer.

### 3.3. Cell Boundary

Here, we set the condition of zero radial flow (no liquid is entering or leaving the cell) and the condition of perfect slip. These conditions apply both in the porous and unstirred layers.
(24)$$u_{\rho }\left(z,R_{c}\right)=0$$
(25)$${\left.\frac{\partial u_{z}}{\partial \rho }+\frac{\partial u_{\rho }}{\partial z}\right|}_{\rho =R_{c}}=0$$
(26)$$v_{\rho }\left(z,R_{c}\right)=0$$
(27)$${\left.\frac{\partial v_{z}}{\partial \rho }+\frac{\partial v_{\rho }}{\partial z}\right|}_{\rho =R_{c}}=0$$

For salt transport, we apply the conditions of zero ion fluxes through the cell surface, which is ensured by zero radial derivatives of the salt concentration and electrostatic potential.
(28)$${\left.\frac{\partial c}{\partial \rho }\right|}_{\rho =R_{c}}=0$$
(29)$${\left.\frac{\partial \varphi }{\partial \rho }\right|}_{\rho =R_{c}}=0$$

### 3.4. IEX Layer

Here, we impose the conditions of zero normal (impermeability) and tangential (zero slip) volume flow
(30)$$u_{z}\left(+0,\rho >R_{h}\right)=0$$
(31)$$u_{\rho }\left(+0,\rho >R_{h}\right)=0$$
(32)$$v_{z}\left(-0,\rho >R_{h}\right)=0$$
(33)$$v_{\rho }\left(-0,\rho >R_{h}\right)=0$$

For ion transport, we use the condition of impermeability to coions
(34)$${\left.\frac{\partial c}{\partial z}+\frac{Fc}{RT}\cdot \frac{\partial \varphi }{\partial z}\right|}_{\begin{array}{c} \rho >R_{h}\\ z=\pm 0 \end{array}}=0$$

For definiteness, we will consider that the IEX layer is impermeable to cations (and, thus, positively charged). Thus, in the discussion below, negative zeta potentials will correspond to the case of fixed charges of opposite signs and positive zeta potentials will correspond to the case of coincident signs. The case of cation-exchange IEX layers (impermeable to anions) can be easily considered by analogy.

In addition to the cation impermeability, we consider that the IEX layer is perfectly permeable to counterions (anions), so their electrochemical potential does not change across the layer
(35)$$ln\left(c\left(+0,\rho <R_{h}\right)\right)-\frac{F}{RT}\varphi \left(c\left(+0,\rho <R_{h}\right)\right)=ln\left(c\left(-0,\rho <R_{h}\right)\right)-\frac{F}{RT}\varphi \left(c\left(-0,\rho <R_{h}\right)\right)$$

Thus, both the salt concentration and electrostatic potential experience jumps at this interface whose magnitudes are related by Equation (35).

### 3.5. Exposed Surface of the Porous Layer within Imperfections

Rigorous hydrodynamic boundary conditions at the interfaces between porous media and free fluids (such as those at the surface of a porous layer exposed within perforations) is a complex theoretical matter, which is additionally complicated by the fact that the Brinkman equation is not rigorous. We resolve this problem by effectively using the Brinkman equation not only in the porous but also in the unstirred layer and by assuming that within the latter, the hydraulic permeability was extremely large. In this way, the Comsol Multiphysics software ensures a correct description of hydrodynamics at the interface between the porous layer and free fluid within perforations without the need to explicitly formulate the boundary conditions.

For the salt transport problem at this boundary, we use the conditions of the continuity of salt concentration and electrostatic potential as well as of flux continuity for each of the two ions, which, taking into account the difference in the diffusion coefficients due to the finite porosity, gives rise to these conditions
(36)$$c\left(z=+0\right)=c\left(z=-0\right)$$
(37)$$\varphi \left(z=+0\right)=\varphi \left(z=-0\right)$$
(38)$${\left.\frac{\partial c}{\partial z}\right|}_{z=-0}=\gamma \cdot {\left.\frac{\partial c}{\partial z}\right|}_{z=+0}$$
(39)$${\left.\frac{\partial \varphi }{\partial z}\right|}_{z=-0}=\gamma \cdot {\left.\frac{\partial \varphi }{\partial z}\right|}_{z=+0}$$

Numerical simulations were performed using Comsol Multiphysics 6.1 software with the 2-D Axisymmetric Geometry Model in the sub-section “Porous Media and Subsurface Flow” of the section “Fluid Flow”, physics interface Brinkman Equation. Close to the perforation and IEX layer, we used adaptive meshes of the type mapped. For the remaining part of the numerical domain, we used a simple free triangular mesh with a maximum element size of around 1 μm in the vertical direction. In the radial direction, the mesh size was adapted to the cell size using the scale geometry option.

## 4. Results and Discussion

Our model has a number of parameters. The most important parameters are the size of perforations, the distance between them, the thickness of the porous layer, the zeta potential of the pore surface and applied voltage. We will explore the dependencies of system performance on the more important of these parameters. The other parameters will be fixed at some reasonable values, as briefly explained below in Table 1.

Due to the deviations from flow one-dimensionality close to the perforations, there are some hydrostatic pressure gradients in the solution beneath the membrane. They can be expected to fade away at some distance from it giving rise to constant hydrostatic pressure and a 1D liquid flow. In auxiliary simulations, we checked the dependence of our results on the location of the plane where hydrostatic pressure is set equal to zero, and we found that there was no dependence at $L_{s}>400÷500\;\mu m$.

The value of hydraulic permeability approximately corresponds to the average size of cylindrical pores of 200 nm. Thus, the pores are much larger than the screening length in the assumed 1 mM KCl solution, so the Smoluchowski approach to the description of electroosmosis is applicable.

Ultimately, the effect of net flow is due to a geometrical asymmetry: a porous layer on one side and an unstirred solution layer on the other. The extent of asymmetry becomes smaller when the unstirred layer becomes thicker. Therefore, one can expect a monotone decrease in the net flow rate with increasing unstirred layer thickness, which was confirmed using simulations (not shown). The value of thickness used in the calculations (50 µm) is typical for systems with moderate stirring.

Our problem is linear in concentration, so only relative changes in concentration matter. The only output property that depends on the salt concentration is the current density (it is directly proportional to it). For definitiveness, we used the value of 1 mM for the initial concentration. For other concentrations, one should adjust the current density proportionally.

The previous study [35] demonstrated that without unstirred layers (and using the Darcy law for the description of flow in the porous layer), the dependencies of flow rate on the applied voltage (including its sign) were strictly linear. The system’s non-linearity and asymmetry manifested themselves only in the current–voltage characteristics. Due to their asymmetry, the same amount of electric charge transferred in two opposite directions gave rise to the transfer of different volumes of liquid and a non-zero net volume transfer over the period. In this study, we account for unstirred layers and use the Brinkman equation (instead of the Darcy law) primarily to better capture the flow details close to the perforation edge. This gives rise to deviations from the strict linearity of flow vs. applied voltage. As one can see from Figure 2, they are not very pronounced in the case of opposite signs of fixed charges in the porous and IEX layers. On the contrary, the current–voltage characteristics in this case are pronouncedly asymmetrical. Therefore, the net flows are controlled by the current asymmetry as described above. In the case of coincident signs of fixed charges, the situation is more complex, namely, both the flow and current feature considerable asymmetry, while the current asymmetry is less pronounced than in the case of opposite signs. However, both asymmetries work in the same direction given that the currents are larger in magnitude at negative voltages where the flow rates are smaller. Below, we will see that in many cases, this gives rise to somewhat larger net flows and efficiencies (see below for the definitions) than in the case of charges of opposite signs, although this ultimately depends on the system parameters. Overall, both configurations feature comparable performances.

Figure 2 shows that the volume flow has a considerable impact on the current–voltage characteristics (especially, at larger voltages). The flow plays a dual role. First, it defines the shape of the quasi-1D salt-concentration profiles away from the interface between the porous and IEX layers. This, in turn, influences the current-induced ICP itself because convection transports salt to or from the polarized interface. Second, the volume transfer causes a solution “injection” or “ejection” through the perforations. Injection denotes flows directed into the porous layer and ejection denotes the oppositely directed flows. The injected solution can have either a higher or lower concentration than the surrounding solution. The injection of a (much) higher concentration has a stronger impact on the concentration in the porous layer (and current density) than the injection of a (much) lower concentration. In turn, the concentration in the injected solution is essentially influenced by the ICP phenomena in the unstirred layer.

As discussed above, in some cases, non-zero net flows (under AC conditions) arise primarily owing to a dependence of current density on the sign of the applied voltage (current direction). Due to such current asymmetry, essentially different amounts of electric charge are transferred in two directions, while the transferred volume is approximately the same. In other words, when the same amount of charge is transferred in two directions (for example, by adjusting the times of current passage in the opposite directions), there is a net volume transfer. Zero net charge transfer is a definition of alternate currents. Their principal advantage is that capacitive electrodes (without electrode reactions) can be used.

The time of passage of a unit charge is inversely proportional to the current. Therefore, the volume transferred in a given direction per unit charge is proportional to the rate of volume transfer divided by the current. The net transfer (per double unit charge) is the difference in volumes transferred in the opposite directions. The time needed to transfer double unit charge (the same unit charge in each direction) is equal to the sum of inverse currents. Therefore, the rate of net volume transfer (the net volume transferred per unit time) is given by the following equation:(40)$$V_{net}\equiv \frac{\frac{V_{1}}{j_{1}}-\frac{V_{2}}{j_{2}}}{\frac{1}{j_{1}}+\frac{1}{j_{2}}}$$
where $V_{1},V_{2}$ and $j_{1},j_{2}$ are the absolute values of volumes and currents transferred in two directions, respectively. Scaled on the per-perforation area, this gives us the linear velocity (m/s) of net volume flow.
(41)$$v_{net}\equiv \frac{1}{\pi R_{c}^{2}}\left(\frac{\frac{V_{1}}{j_{1}}-\frac{V_{2}}{j_{2}}}{\frac{1}{j_{1}}+\frac{1}{j_{2}}}\right)$$

This will be one of the principal output values considered below. Of course, this definition is also valid in the cases where the flow rate is significantly asymmetrical.

Adjusting the times of current passage in two directions is not the only possible scenario. For example, one could also apply different (in absolute value) voltages of opposite signs for approximately the same times. Nonetheless, for definitiveness, in this study, we will use this simple definition of net flow rate. Selecting an optimal protocol is the subject matter of application-specific optimization, which is beyond the scope of this study.

The numerator in Equation (40) is the net volume transferred per double unit charge (a unit charge is transferred in each direction). In addition to its scaling by the time of transfer for this charge (as in Equation (40)), we can normalize it by the volume moved as a result of the transfer of the same charge in conventional electro-osmosis with the same porous material but without any perforated IEX mask. Within the scope of our model, this is described by the simple Smoluchowski formula for the linear velocity of electro-osmosis
(42)$$J_{v}=\frac{8\varepsilon \varepsilon_{0}\gamma \zeta }{\eta }\frac{\Delta \varphi }{L}$$

The current density in this simple case is
(43)$$I\equiv \gamma \lambda \frac{\Delta \varphi }{L}$$
where λ is the bulk solution conductivity (we neglect surface conductance). Taking the ratio, we obtain $8\varepsilon \varepsilon_{0}\zeta /\eta \lambda$ for the volume transferred per unit charge. Scaling the net volume transferred per unit charge (half of the numerator in Equation (40)) by this value produces a dimensionless parameter that informs us about the efficiency of the AC EO pump as compared to the maximum possible performance.
(44)$$\Theta \equiv \frac{\eta \lambda }{16\varepsilon \varepsilon_{0}\zeta }\left(\frac{V_{1}}{j_{1}}-\frac{V_{2}}{j_{2}}\right)$$

This will be another principal output property described below. We will see that some trends in its dependencies on the system parameters can be different from those observed for the linear velocity of the net volume flow defined by Equation (41). Accordingly, optimal combinations of parameters can be different depending on whether the net flow rate or the volume transferred per a given charge (for example, of a battery) is to be maximized.

The direction of electro-osmosis is towards the movement of counterions. In the current-induced concentration polarization of interfaces between electrolyte solutions and IEX media, ion depletion (salt-concentration reduction) occurs at interfaces receiving counterions from the solution. In the opposite case (interfaces releasing counterions), there is ion enrichment (salt-concentration increase). Thus, in our system with two “active” (EO-active and perm-selective IEX) layers, there are four possible configurations:Concentration decreased in the porous layer, and flow away from the interface (Figure 3a).Concentration increased in the porous layer, and flow towards the interface (Figure 3b).Concentration increased in the porous layer, and flow away from the interface (Figure 3c).Concentration decreased in the porous layer, and flow towards the interface (Figure 3d).

The first two configurations (a,b) correspond to opposite signs of fixed charges in the IEX and porous layers. The last two configurations (c,d) correspond to coincident signs.

At typical parameter combinations, the electrical resistance of the porous layer is dominant and controls the current density at a given applied voltage. This resistance is controlled by the salt concentration, so its distribution in the porous layer is of primary importance. This concentration is controlled by two factors. First, the concentration changes induced by the current passage across the interface between the IEX and porous layers. Second, the convective “injection” of a solution through perforations from the nearby unstirred layer. The concentration in this “injected” solution is always changed in the opposite direction from the porous layer (i.e., increased when in the porous layer it is decreased and vice versa). Whether there is an injection into or ejection from the porous layer depends on the relationship between the current direction and the sign of the zeta potential of the pore surface. In turn, depending on the relationship between the sign of perm-selectivity of the IEX layer and the current direction, the injected solution can have either increased or reduced concentration. Under the strongly non-linear conditions of this study, these decreases or increases can be quite strong. It is clear that the injection of a high-concentration solution into a dilute one has a much stronger effect on the resulting (after a diffusive mixing) concentration in the porous layer than the injection of a dilute solution into a concentrated one.

Of course, this picture is oversimplified because the current density is far from homogeneous and very strongly depends on the radial position. This current inhomogeneity can be expected to be more pronounced in the configuration where a more concentrated solution is injected into the porous layer (Figure 3a) because this creates a zone of high conductivity “focusing” current streamlines (see Figure 4). This zone is connected in parallel to the remaining part having much lower conductivity. Of course, this inhomogeneity is progressively leveled out by diffusion away from the interface.

Below, we will see that the configuration with fixed charges of the same sign in the porous and IEX layers overall seems to show a better performance, especially in terms of efficiency. This may be related to a better radial homogeneity in the concentration (and current) distribution in this case.

In this system, perforations in the IEX layer are a central element enabling fluid transfer. At the same time, their properties such as size and spacing seem to be relatively easy to engineer. There are two obvious limiting cases, namely, a very large and a very small spacing. In both cases, the net velocity of the volume flow should tend to zero. At very large spacing, this occurs because a finite volume flow through a perforation is distributed over an ever larger per-perforation area. At very small spacing, the perforations practically overlap, so there is almost no IEX layer and no associated current-induced ICP. The flow rates in both directions are large but strictly symmetrical, so the net volume flow is again zero. From these limiting cases, it follows that the dependencies of the net flow velocity on the inter-perforation spacing should have maxima. This is confirmed by Figure 5 showing the velocity of the net volume flow and efficiency as functions of the cell radius (directly related to the inter-perforation spacing) for various perforation radii.

As expected, all the dependencies have maxima. The most important observation is that the height of these maxima does not decrease very much even when the perforation size is increased by almost two orders of magnitude. Moreover, the optimal cell radii (corresponding to the maxima) increase with the perforation size. In practical terms, this is good news because larger perforations at a larger spacing are definitely easier to generate. Indeed, the optimal spacing increases sub-linearly with the perforation size, so optimal porosity increases, for example, from about 0.7% in the case of $R_{h}=1\;\mu m$ to around 21% for $R_{h}=64\;\mu m$.

All the curves in Figure 5 look qualitatively similar. Nevertheless, there are some potentially important quantitative differences. Thus, for instance, the maxima at small perforation sizes are somewhat higher in the case of coincident signs of fixed charges, but their height decreases with increasing perforation size faster than in the case of opposite signs. Therefore, coincident signs can be preferable for smaller perforations, and systems with different signs could work better for larger perforations. In terms of efficiency, the trends are similar, but the advantage of coincident signs at smaller perforation sizes is more pronounced. Finally, somewhat larger spacings are optimal in terms of efficiency than in terms of the velocity of the net volume flow.

The dependencies shown in Figure 5 were calculated for a relatively large applied voltage magnitude (600 mV). Figure 6 shows similar dependencies obtained for a lower voltage magnitude of 200 mV (they also enable easier comparison of cases of opposite and coincident signs). Although the dependencies again are qualitatively similar, the most interesting observation is that the efficiency at the lower voltage is noticeably smaller, especially for the configuration with coincident signs of fixed charges in the porous and IEX layers. Thus, for instance, at the point of maximum for $R_{h}=1\;\mu m$, the efficiency is about 0.107 at $\varphi_{0}=600\;\mathrm{mV}$ but only around 0.067 at $\varphi_{0}=200\;\mathrm{mV}$. A close inspection of Figure 5 and Figure 6 reveals that the net flow velocity at the points of maxima also increases somewhat super-linearly with the applied voltage.

This, however, concerns the net velocities and efficiencies at the points of maxima. Their location, in turn, depends on the applied voltage. Figure 7 shows some dependencies on applied voltage calculated for a fixed combination of perforation and cell sizes.

The dependencies of net velocities are initially super-linear and then become sublinear, so the benefits of increasing voltage depend on its range. The efficiency has maxima located differently depending on the sign and magnitude of the zeta potential. The coincident signs of fixed charges are always beneficial in terms of efficiency but are not always so in terms of the net flow velocity. In summary, the applied voltage is another parameter to be selected on a case-by-case basis depending on the application and optimization criteria. One should also keep in mind that excessively large voltages can give rise to electrode reactions (e.g., water splitting) and, thus, compromise the capacitive function of the electrodes.

The porous layer thickness can probably also be engineered with relative ease. The dependencies of net flow velocity on the thickness can be expected to have maxima because too thin layers have insufficient “internal” hydraulic resistance to be able to pump liquid through the entrance resistance of the perforated IEX layer, while in excessively thick layers, the driving voltage gradients are reduced. In terms of efficiency, the situation can be different because increasing thickness gives rise to decreasing currents. Figure 8 shows that, indeed, the dependencies of net velocity on the porous layer thickness have pronounced maxima at H=60÷80 μm, while the maxima for efficiency are much broader and shifted towards larger thicknesses. However, the location of the maxima depends on a number of parameters and has to be determined on a case-by-case basis. The value we used in most of the above simulations (H=100 μm) is a compromise between the maximum net velocities and efficiencies. The efficiency increases with the thickness within its broader range. Nevertheless, it also decreases starting from certain thicknesses, which is probably caused by a dependence on the thickness of ICP phenomena.

The zeta potential of the pore surface is more difficult to engineer than the “geometrical” parameters, but this can still be possible to some extent using material selection or modification. Generally, convective flows (controlled by the zeta potential) seem to always reduce the ICP and, thus, the system asymmetry. However, both direct and reverse flows increase with the magnitude of the zeta potential. Within a certain range, this overcompensates the loses in the asymmetry, and the net flow velocity increases. Figure 9 shows that this occurs only up to certain values of the zeta potential depending on its sign and the geometrical parameters. Remarkably, efficiency is highest at very low zeta potentials. This is one more example of the classical trade-off between efficiency and productivity. Again, the optimal values should be selected considering application-specific optimization criteria.

## 5. Conclusions and Outlook

Using numerical simulations, we have explored a novel electro-membrane microfluidic diode featuring significant (to be of practical interest) net volume flows in response to (ultra)low-frequency AC voltages. The system consists of a nanoporous layer and a micro-perforated ion-exchange layer put in series. The dependencies of system performance on model parameters such as the perforation size and spacing, the porous layer thickness and the zeta potential of pore surface feature maxima offer opportunities for system optimization for practical applications, for example, in sports garments with “active” moisture evacuation. In particular, our simulations have revealed that even relatively large perforations (~100 μm) can give rise to a decent performance, which is important in terms of practical implementability and cost. In addition, we have demonstrated that configurations with coincident signs of fixed charges in the ion-exchange and porous layers can feature even better performance than systems with opposite signs as considered previously. This is of interest in view of the typically better properties of cation-exchange materials (as compared to anion-exchange ones) and reduced fouling of negatively-charged surfaces by natural organic matter (as compared to positively-charged ones).

In this study, we considered only stationary solutions, which implies that the electrical capacity of electrodes was sufficiently large to ensure practically constant voltage drops in the system over times that are much longer than the characteristic relaxation times of concentration changes. This assumption is probably unrealistic, so for analysis of practical systems, non-stationary simulations will be needed. Explicit inclusion of electrodes will require even more model parameters than used in this study and will have to be performed in conjunction with experimental studies where the values of a part of parameters can be fixed according to specific systems.

## Figures and Tables

**Figure 1 membranes-13-00243-f001:**
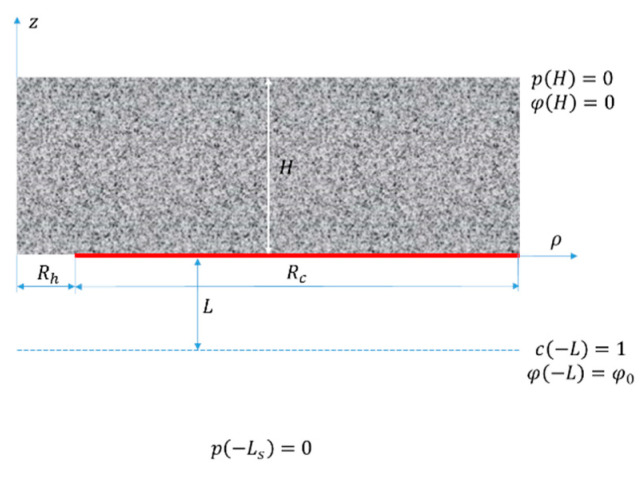
The model geometry and system of coordinates; the red line shows the selective layer.

**Figure 2 membranes-13-00243-f002:**
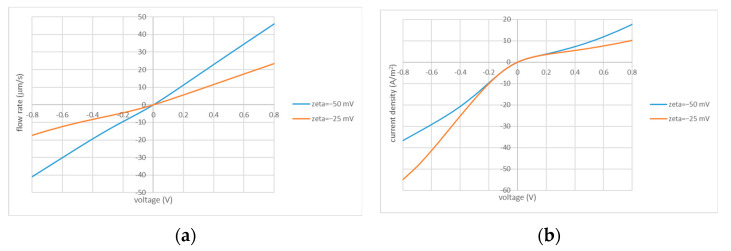
Current–voltage (**b**,**d**) and flow–voltage (**a**,**c**) characteristics: $R_{h}=1\;\mu m$; $R_{c}=10\;\mu m$; the values of the zeta potential are indicated in the legends.

**Figure 3 membranes-13-00243-f003:**
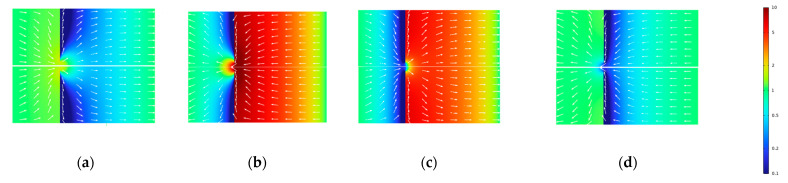
Schematic of the concentration distribution (color) and flow direction (arrows): $R_{h}=4\;\mu m$; $R_{c}=60\;\mu m$; (**a**) $\varphi_{0}=0.6\;V;\;\zeta =-50\;\mathrm{mV}$; (**b**) $\varphi_{0}=-0.6\;V;\;\zeta =-50\;\mathrm{mV}$; (**c**)$\;\varphi_{0}=-0.6\;V;\;\zeta =50\;\mathrm{mV}$; and (**d**) $\varphi_{0}=0.6\;V;\;\zeta =50\;\mathrm{mV}$.

**Figure 4 membranes-13-00243-f004:**
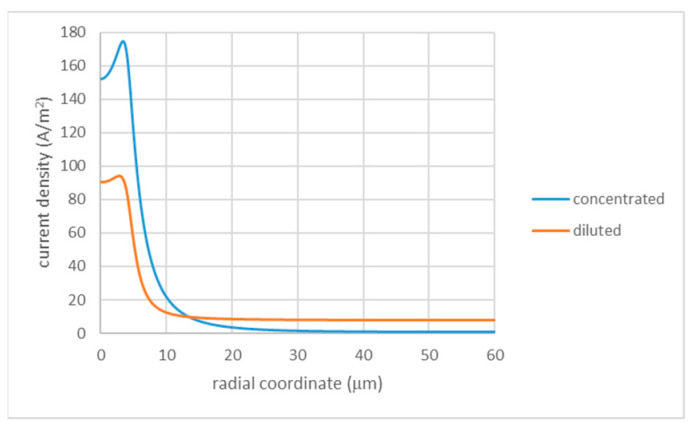
Radial distribution of current density in configurations with solution injection: $R_{h}=4\;\mu m$; $R_{c}=60\;\mu m$; $\varphi_{0}=0.6\;V;\;\zeta =-50\;\mathrm{mV}$ (injection of a more concentrated solution); $\varphi_{0}=-0.6\;V;\;\mathrm{and}\;\zeta =50\;\mathrm{mV}$ (injection of a more diluted solution). The legend indicates the condition of injected solution.

**Figure 5 membranes-13-00243-f005:**
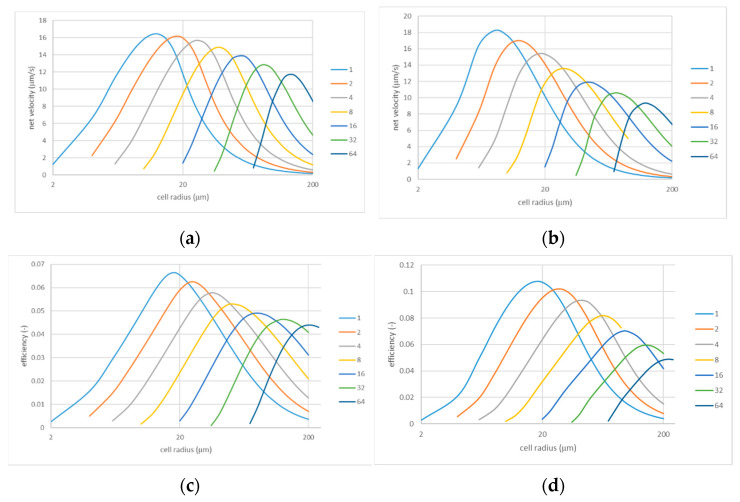
Net flow velocity and efficiency vs. cell radius for various perforation sizes: $\varphi_{0}=0.6\;V$; ζ=−50 mV (**a**,**c**) and ζ=50 mV (**b**,**d**). The legends indicate the perforation radii (in µm); note that $1\;\mu m/s\equiv 3.6\;l/\left(m^{2}h\right)$.

**Figure 6 membranes-13-00243-f006:**
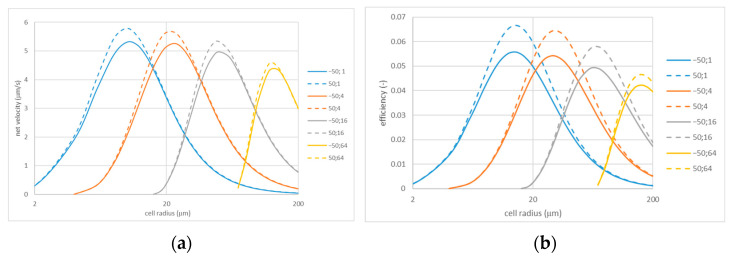
Net flow velocity (**a**) and efficiency (**b**) vs. cell radius for various perforation sizes: $\varphi_{0}=0.2\;V$; the legends indicate zeta potentials (in mV) and perforation radii (in µm). Note that the solid lines show the case of opposite signs of fixed charges and the dashed lines show the case of coincident signs.

**Figure 7 membranes-13-00243-f007:**
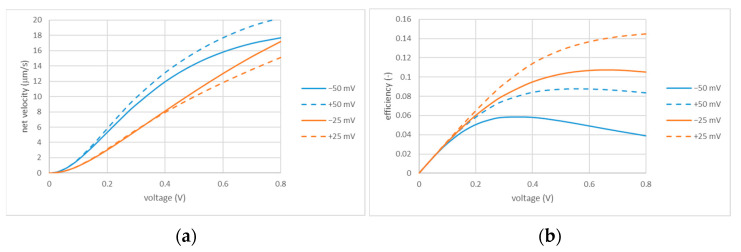
Net flow velocity (**a**) and efficiency (**b**) vs. applied voltage; $R_{h}=1\;\mu m$ and $R_{c}=10\;\mu m$. The values of zeta potential are indicated in the legends.

**Figure 8 membranes-13-00243-f008:**
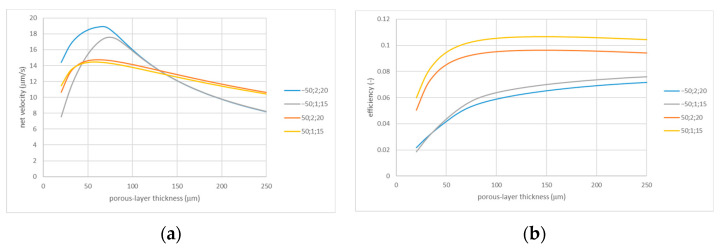
Net flow velocity (**a**) and efficiency (**b**) vs. porous layer thickness; the legends indicate zeta potential (mV); imperfection radius (µm); and cell radius (µm).

**Figure 9 membranes-13-00243-f009:**
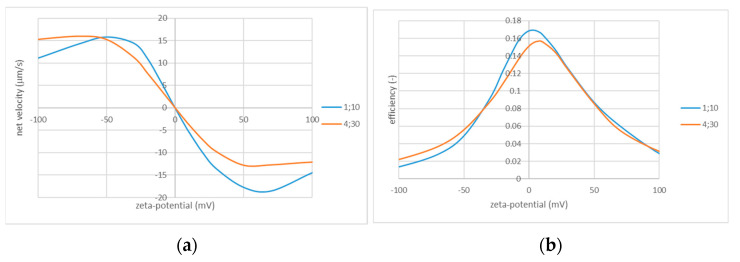
Net flow velocity (**a**) and efficiency (**b**) vs. zeta potential: perforation radius (in µm) and cell radius (in µm) are indicated in the legend.

**Table 1 membranes-13-00243-t001:** Model parameters.

Parameter	Notation	Value(s)
relative dielectric constant	ε	80
temperature	T	300 K
salt (ion) diffusion coefficient	D	$2\cdot 10^{-9}m^{2}/s$
viscosity	η	1 cP
salt concentration	c	1 mM
porous layer thickness	*H*	100 μm (variable in Figure 8)
unstirred layer thickness	L	50 μm
constant pressure distance	$L_{s}$	500 μm
porous layer porosity	γ	0.3
porous layer hydraulic permeability	k	$1.2\cdot 10^{-13}m^{2}/s\cdot \mathrm{Pa}$
perforation radius	$R_{h}$	1÷64 μm
cell radius	$R_{c}$	2÷200 μm
zeta potential	ζ	±50 mV(variable in Figure 9)

## Data Availability

Data available on request.

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
