# Peer review of "Computational Design of an Electro-Membrane Microfluidic-Diode System"

_membranes, 2023, doi:10.3390/membranes13020243_

Round 1
Reviewer 1 Report
The "membranes-2229258" manuscript presents a mathematical study of microfluidic-diode system. There is no doubt that the area of the presented work is relevant. The manuscript is well and logically structured. I have just a few comments to improve understanding.
I don't fully agree with the statement "…Only few studies actually also studied the associated volume transfer, for example, in nano-funnels [18], or track-etched membranes with conical nano-pores [19– 30]…". Some of the effects described by the authors in the introduction have been studied in many theoretical papers. In particular, this was mentioned in recent papers by A. Filippov (10.3390/ijms232112778; 10.3390/colloids6020034), as well as by V. Nikonenko's group (in particular, in the recent ones 10.3390/ijms24010034; 10.3390/membranes12121283).
The paper doesn't describe in detail the current mode used in this model. Only one place seems to mention that the AC condition is used. I understand that the model does not use time dependencies, but the main sections should emphasize the way of working with the simulated system of porous layers.
Author Response
Comment: I don't fully agree with the statement "…Only few studies actually also studied the associated volume transfer, for example, in nano-funnels [18], or track-etched membranes with conical nano-pores [19– 30]…". Some of the effects described by the authors in the introduction have been studied in many theoretical papers. In particular, this was mentioned in recent papers by A. Filippov (10.3390/ijms232112778; 10.3390/colloids6020034), as well as by V. Nikonenko's group (in particular, in the recent ones 10.3390/ijms24010034; 10.3390/membranes12121283).
Reply: There are numerous studies (not only those mentioned by the Reviewer) considering simultaneous ion and water transfer, in particular, in electro-membrane systems. However, as it clearly follows from the previous sentence, our statement concerns only studies devoted to nanofluidic diodes.
Comment: The paper doesn't describe in detail the current mode used in this model. Only one place seems to mention that the AC condition is used. I understand that the model does not use time dependencies, but the main sections should emphasize the way of working with the simulated system of porous layers.
Reply: We are grateful to the Reviewer for this very pertinent comment. The use of stationary solutions for the modelling of an AC EO pump is explained in the Conclusions but, indeed, this definitely had to be done before. Now, corresponding explanations are provided at the end of Introduction section.

Reviewer 2 Report
In my opinion, this is a good study. The structure of the paper is complete, the logic is clear, The paper can be published after minor revise.
1. Line 26, it is unnecessary to put 17 references in one place.
2. The reason of “The most important observation is 427 that the height of these maxima does not decrease very much even when the perforation 428 size is increased by almost two orders of magnitude.” should be given.
3. The font format in Figure 6 is different.
Author Response
Comment: Line 26, it is unnecessary to put 17 references in one place.
Reply: By providing a relatively large number of references, we wanted to stress the high publication activity in this field.
Comment: The reason of “The most important observation is 427 that the height of these maxima does not decrease very much even when the perforation 428 size is increased by almost two orders of magnitude.” should be given.
Reply: The trends observed in this study are results of combination of complex coupled strongly non-linear phenomena. In some cases, we could find relatively simple qualitative explanations but in other cases this was not possible. In fact, just this makes the numerical analysis really useful.
Comment: The font format in Figure 6 is different.
Reply: Sorry, we don´t understand this comment.